# Reducing Immunosuppression in Patients with De Novo Lung Carcinoma after Liver Transplantation Could Significantly Prolong Survival

**DOI:** 10.3390/cancers14112748

**Published:** 2022-06-01

**Authors:** Sina Pesthy, Elisa Wegener, Ramin Raul Ossami Saidy, Lea Timmermann, Deniz Uluk, Mustafa Aydin, Tomasz Dziodzio, Wenzel Schoening, Georg Lurje, Robert Öllinger, Nikolaj Frost, Uli Fehrenbach, Jens-Carsten Rückert, Jens Neudecker, Johann Pratschke, Dennis Eurich

**Affiliations:** 1Department of Surgery, Campus Virchow-Klinikum and Campus Charité Mitte, Charité—Universitätsmedizin Berlin, Augustenburger Platz 1, 13353 Berlin, Germany; elisa.wegener@charite.de (E.W.); ramin-raul.ossami-saidy@charite.de (R.R.O.S.); lea.timmermann@charite.de (L.T.); deniz.uluk@charite.de (D.U.); mustafa.aydin@charite.de (M.A.); tomasz.dziodzio@charite.de (T.D.); wenzel.schoening@charite.de (W.S.); georg.lurje@charite.de (G.L.); robert.oellinger@charite.de (R.Ö.); jens-c.rueckert@charite.de (J.-C.R.); jens.neudecker@charite.de (J.N.); johann.pratschke@charite.de (J.P.); dennis.eurich@charite.de (D.E.); 2BIH Charité Clinician Scientist Program, Berlin Institute of Health at Charité—Universitätsmedizin Berlin, Charitéplatz 1, 10117 Berlin, Germany; 3Department of Infectious Diseases and Pulmonary Medicine, Campus Virchow-Klinikum and Campus Charité Mitte, Charité—Universitätsmedizin Berlin, Augustenburger Platz 1, 13353 Berlin, Germany; nikolaj.frost@charite.de; 4Department of Radiology, Campus Virchow-Klinikum and Campus Charité Mitte, Charité—Universitätsmedizin Berlin, Augustenburger Platz 1, 13353 Berlin, Germany; uli.fehrenbach@charite.de

**Keywords:** liver transplantation, de novo lung cancer, lung carcinoma, immunosuppression, surgical tumor resection

## Abstract

**Simple Summary:**

Long-term immunosuppressive therapy following liver transplantation is associated with an increased risk for the development of de novo lung carcinoma. However, data on the management of the immunosuppression following the diagnosis of lung cancer are missing to the present day. In this retrospective analysis, we investigate factors associated with improved survival of liver transplant recipients with diagnosis of de novo lung carcinoma with a particular emphasis on the impact of immunosuppression. Our findings suggest that strict reduction of immunosuppression has a beneficial effect on survival in this particular situation and, thus, should be an early intervention following diagnosis. Liver transplant recipients with the diagnosis of de novo lung cancer should be offered surgical treatment if technically feasible as a potential curative therapeutic option to improve limited prognosis. Further investigations concerning dosage findings and reduction of immunosuppression in organ recipients should be the targets of subsequent studies.

**Abstract:**

(1) Background: Liver transplantation (LT) is an established treatment for selected patients with end-stage liver disease resulting in a subsequent need for long-term immunosuppressive therapy. With cumulative exposure to immunosuppression (IS), the risk for the development of de novo lung carcinoma increases. Due to limited therapy options and prognosis after diagnosis of lung cancer, the question of the mode and extent of IS in this particular situation is raised. (2) Methods: All patients diagnosed with de novo lung cancer in the follow-up after LT were identified from the institution’s register of liver allograft recipients (Charité—Universitätsmedizin Berlin, Germany) transplanted between 1988 and 2021. Survival analysis was performed based on the IS therapy following diagnosis of lung cancer and the oncological treatment approach. (3) Results: Among 3207 adult LTs performed in 2644 patients at our institution, 62 patients (2.3%) developed de novo lung carcinoma following LT. Lung cancer was diagnosed at a median interval of 9.7 years after LT (range 0.7–27.0 years). Median survival after diagnosis of lung carcinoma was 13.2 months (range 0–196 months). Surgical approach with curative intent significantly prolonged survival rates compared to palliative treatment (median 67.4 months vs. 6.4 months). Reduction of IS facilitated a significant improvement in survival (median 38.6 months vs. 6.7 months). In six patients (9.7%) complete IS weaning was achieved with unimpaired liver allograft function. (4) Conclusion: Reduction of IS therapy after the diagnosis of de novo lung cancer in LT patients is associated with prolonged survival. The risk of acute rejection does not appear to be increased with restrictive IS management. Therefore, strict reduction of IS should be an early intervention following diagnosis. In addition, surgical resection should be attempted, if technically feasible and oncologically meaningful.

## 1. Introduction

Within the last decades, liver transplantation (LT) has become an established treatment procedure for selected patients with end-stage liver disease. The improvement in survival of transplant recipients is attributed to progress in surgical techniques and perioperative care as well as the administration of post-transplant immunosuppression (IS) which significantly reduced the risk of acute graft rejection [1,2]. Over 7000 LTs are performed annually in Europe resulting in a continuously expanding number of transplant recipients with the need for long-term IS [3]. Calcineurin inhibitors (CNI), mycophenolate mofetil (MMF), and mammalian target of rapamycin inhibitors (mTORi) are most frequently used as standard IS [2].

With prolonged survival and hence cumulative exposure to IS, multiple systemic complications of long-term course of IS therapy emerge progressively [4,5]. Infections, de novo malignancies, tumor recurrence, and cardiovascular events are directly associated with IS intake [5,6,7,8]. One of the most severe complications of LT and its subsequent IS therapy is the increased risk for the development of de novo malignancies due to impairment in immunosurveillance [9,10]. Several studies show that with cumulative duration and intensity of therapeutic IS, the risk of various malignant tumors increases to 2.6–4.3-fold compared to the general population [5,11,12]. In the case of diagnosis of a de novo malignancy, prognosis and therapeutic options are usually limited leading to significantly reduced survival rates. In fact, post-transplant malignancies are the second leading cause of death among adult organ recipients [7,9]. The association between standard IS protocols after LT and posttransplant carcinogenesis justifies further research on how to proceed with IS after diagnosis of malignancy.

Based on the potential antiproliferative effect of mTORi, the most common tendency is to change the basis of IS to mTORi in combination with low-dose tacrolimus but without reliable evidence [1,2,4,13]. Strict reduction of IS in recipients with preserved graft function is another first-line approach that is discussed to positively affect survival of patients after diagnosis of HCC recurrence following LT [14].

Referring exclusively to de novo lung carcinoma, long-term survival of LT recipients is threatened by two- and three-fold higher incidence of lung cancer in comparison with the general population [15]. In a study on almost 90,000 LT recipients, lung carcinoma accounted for 26% of de novo malignancy-related deaths following LT [16]. However, further data on the mode and extent of IS therapy for prevention of de novo lung cancer and for improvement of prognosis after diagnosis of lung carcinoma are still missing. Based on our findings, we hypothesize that reduction of IS to the point of discontinuation might serve as a feasible oncological co-treatment.

The aim of this retrospective study was to provide evidence from our 30 years lasting LT experience and to investigate factors associated with improved survival of LT recipients after diagnosis of de novo lung carcinoma after LT with a particular emphasis on the impact of IS.

## 2. Materials and Methods

This retrospective study was performed according to the guidelines of the Declaration of Helsinki and was approved by the local Ethics Committee of Charité—Universitätsmedizin Berlin (protocol code EA1/035/21, date of approval 18 February 2021).

In total, 3207 adult LT in 2644 patients were performed for various end-stage liver diseases between 1988 and 2021 at our transplant center (Department of Surgery, Charité—Universitätsmedizin Berlin).

Follow-up data were acquired by in-hospital data and reports from external institutions including primary care physicians, local gastroenterologists and oncologists. Data on clinical, laboratory, and histological parameters were extracted from a prospectively maintained database and evaluated retrospectively.

In a life-long surveillance concept, all transplant recipients were followed-up both clinically and radiographically on a regular basis at our outpatient center. Serological tests were based on the time after LT and ranged between 2x/week to every twelve weeks. To detect or exclude typical pathologies clinical, biochemical, and histological examinations as well as abdominal ultrasound were performed routinely according to our standard protocol at 1, 3, 5, 7, 10, 13, 15, 17, 20, 23, and 25 years and so on after LT. Chest X-ray was performed to detect pulmonal malignancies, accompanied by computed tomography (CT) of the chest with biopsy in case of conspicuous results or bronchoscopy with biopsy in an individual manner.

In case of detected pulmonary carcinoma, de novo lung cancer was dichotomized into non-small cell lung cancer (NSCLC) or small cell lung cancer (SLC) based on histopathological findings. Pulmonary metastases from other entities (*n* = 2) were excluded from the analysis. NSCLC was subdivided into squamous cell carcinoma and adenocarcinoma.

Therapeutic options were discussed depending on the extent of the oncological disease in a multidisciplinary tumor board. Best supportive care (BSC), chemotherapy, radiotherapy or surgical resection were assessed.

To analyze the impact of IS on the survival of LT recipients after diagnosis of pulmonary malignancy after LT, IS regimen and dosage was reevaluated. All patients were grouped into three categories: Reduction of CNI, consistency of CNI and withdrawal of CNI. Conversion to mTORi therapy was also assessed.

Statistical analysis was performed using SPSS Statistics Version 28.0 (IBM Co., Armonk, NY, USA). Univariate analysis and Kaplan–Meier analysis were used for comparison and illustration of survival differences and calculation of both Breslow- and log rank-test were conducted to evaluate the short- and long-term effect. Both multivariate and univariate Cox-regression-models were used to analyze effect strength, with a hazard ratio (HR) of less than one indicating survival benefit. A *p*-value of <0.05 was considered significant.

## 3. Results

In total, 62 LT recipients (*n* = 62; 2.3%) developed de novo lung cancer. Follow-up data were assessed in accordance with our standard protocol until November 2021. At this point, 22.6% (*n* = 14) of the study population were still alive.

The clinical characteristics of patients with development of posttransplant pulmonary malignancy were analyzed (Table 1). In accordance with male preponderance in the LT cohort, the majority of the study population was male (*n* = 36; 58.1%). Median age at the time of transplantation was 53.5 years (range 31.3–65.5 years). The main indication for LT were alcoholic liver cirrhosis (*n* = 29; 46.8%), followed by virus-related liver cirrhosis (*n* = 22; 35.5%)—hepatitis C virus-related cirrhosis (*n* = 10; 16.1%) and hepatitis B virus-related cirrhosis (*n* = 12; 19.4%), as well as liver disease with an autoimmune pathogenesis (*n* = 6; 9.7%) such as primary biliary cirrhosis (PBC), primary sclerosing cholangitis (PSC), and autoimmune hepatitis (AIH). Hepatocellular carcinoma (HCC) was diagnosed in 12 LT recipients (19.4%). Retransplantation prior to diagnosis of de novo malignancy was documented in six patients (9.7%) due to loss of graft function. The leading cause for retransplantation was hepatic artery thrombosis (*n* = 3; 50.0%), followed by initial non-function (INF) (*n* = 2; 33.3%) and vena cava thrombosis (*n* = 1; 16.7%).

De novo pulmonary carcinoma was diagnosed at a median follow up of 9.7 years after LT (range 0.7–27.0 years). Regarding the interval from LT to development of de novo pulmonary malignancy, lung cancer occurred in almost one third of patients (*n* = 20; 32.3%) within 5 to 10 years, in 18 patients (29%) between 10 and 15 years, and in 13 patients (20.3%) within 5 years after LT (Figure 1). Five patients (8.1%) developed de novo lung carcinoma 15 to 20 years following LT and three patients (4.8%) each between 20 to 25 years and beyond 25 years after LT.

The overall median survival was 13.2 months (range 0–196 months). Subgroup analysis showed that median survival was 10.6 months (range 0–92 months) in patients with the diagnosis of lung carcinoma 5 years from transplant and 20.7 months each when lung cancer was diagnosed 5–10 years and 10–15 years after LT. With 6.5 months (range 2–54 months) median survival was lowest in patients with the diagnosis of lung carcinoma 15 to 20 years after LT.

Regarding histopathological criteria, NSCLC was identified as the most common entity (*n* = 53; 85.5%). Subgroup analysis of NSCLC showed that 50.9% of NSCLC (*n* = 27) were assigned to squamous cell carcinoma, whereas adenocarcinoma was present in 45.3% of NSCLC (*n* = 24).

All 14 patients who reached the last point of follow-up were diagnosed with NSCLC (Table 2) with equal distribution between squamous cell carcinoma and adenocarcinoma (*n* = 7 each; 50% each).

With reference to the deceased patients (Table 2), NSCLC was present in 81.3% of patients (*n* = 39), whereas adenocarcinoma was diagnosed with reduced frequency (*n* = 5; 10.4%). Among deceased patients with prior diagnosis of NSCLC (*n* = 39), squamous cell carcinoma occurred in 51.3% of cases (*n* = 20) while 43.6% of patients developed adenocarcinoma.

De novo lung carcinoma was detected at early stage (stage I) in 32.3% of all cases (Table 2). Stage II was identified in four patients (6.4%), six patients (9.7%) presented with stage III and seven patients (11.3%) were diagnosed with stage IV carcinoma. In 40.3% of cases exact staging could not be performed at our institution due to the retrospective character of the study and patients who received diagnosis and treatment in an external clinic.

Early recognition was achieved in the majority of patients who were still alive at the last time of follow-up (*n* = 9 of 14, 64.3%). Only 7.1% of patients (*n* = 1 of 14) showed advanced stages at the point of diagnosis. With reference to deceased patients 12.5% (*n* = 6 of 48) were diagnosed with advanced carcinoma. In conclusion, 22.9% of patients (*n* = 11) presented with stage I, 6.3% (*n* = 3) with stage II and stage III was present in 8.4% of patients (*n* = 4). Half of the data referring the tumor stage (*n* = 24; 50.0%) were not available.

Therapeutic approaches after diagnosis of de novo lung carcinoma were assessed. The minority of patients (*n* = 13; 21.0%) received solely BSC due to advanced disease or reduced physical status. In the Kaplan–Meier analysis, long-term survival was significantly impaired in patients treated with BSC alone compared to patients who underwent chemotherapy or surgical resection (Figure 2 median survival 2.5 months (confidence interval (CI): 1.5–3.5; interquartile range (IQR) 0.5 vs. 22.4 months (CI: 2.2–42.6; IQR 10.3), respectively; log rank *p* < 0.001; Breslow *p* < 0.0001).

Systemic chemotherapy and local radiotherapy were initiated in 24 patients (38.7%). Surgical approach was performed in 25 patients after diagnosis of lung carcinoma, of which 19 patients (30.6%) underwent surgical resection and six patients qualified for multimodal therapy (9.7%).

Surgical approach with curative intent facilitated a significant improvement in short- and long-term survival compared to non-surgical treatment including BSC without additional interventions or systemic chemotherapy or radiotherapy (Figure 3 median survival 67.4 months (CI: 28.6–106.2; IQR 19.8) vs. 6.4 months (CI: 2.8–10.0, IQR 1.8), respectively; log rank *p* < 0.0001; Breslow *p* < 0.0001).

At last time of follow-up, 14 patients (22.6%) were still alive (Table 2). Analysis of these patients revealed that 92.9% (*n* = 13) were treated with curative intent. Among those, the majority of patients were treated with a surgical approach (*n* = 8; 57.1%), whereas 21.4% of patients received systemic chemotherapy (*n* = 3) and two patients qualified for a multimodal treatment combining surgery and systemic therapy (14.3%). Only one patient received solely BSC. Interestingly, the administration of IS was stopped.

Further analysis of the 48 deceased patients showed that BSC was performed in 25% of patients (*n* = 12) according to their palliative situation. Regarding those 36 patients receiving further treatment, the majority of patients (*n* = 21; 43.8%) were treated with systemic chemotherapy. Only 22.9% of patients qualified for surgical resection (*n* = 11) and 8.3% of patients (*n* = 4) underwent a multimodal treatment.

At time of diagnosis of lung carcinoma, CNIs alone or in combination were used in 60 patients (97%). Among those patients, tacrolimus was part of IS therapy in 48 patients (77.4%) with a mean dosage of 2.6 mg per day and median trough level of 5.2 ng/mL (range 1.4–10.5 ng/mL). In patients with tacrolimus monotherapy (*n* = 30; 48.5%), tacrolimus was administered with a mean dosage of 3.0 mg per day and median trough level of 5.4 ng/mL (range 1.8–10.5 ng/mL). Dual treatment with tacrolimus in combination with MMF was the second most common IS regimen (*n* = 15; 24.2%) with mean tacrolimus dosage of 2.1 mg per day and a median trough level of 5.2 ng/mL (range 1.4–9.4 ng/mL) and mean MMF dosage 1.1 g per day and a median MMF trough level of 1.5 µg/mL (range 0.4–4.4 µg/mL). IS therapy was based on cyclosporin A (CYA) in 12 patients (19.4%) at time of tumor diagnosis with a mean dosage of 234.4 mg per day and median trough level of 266 μg/L (range 63.2–788.0 μg/L).

Modification of IS therapy following diagnosis of de novo lung cancer was assessed (Table 2). Diagnosis of lung carcinoma initiated a reduction of IS therapy in the majority of patients (*n* = 33; 53.2%). CNI was reduced from a mean dosage of 2.6 mg per day and median trough level of 5.2 ng/mL at the time of lung carcinoma diagnosis to 1.5 mg per day and median trough level of 2.2 ng/mL (range 0.0–8.9 ng/mL) after diagnosis of lung cancer. Analysis for paired testing found this reduction to be statistically significant (*p <* 0.001).

However, in 23 patients (37.1%) IS was not modified. Six patients (9.7%) were weaned off IS successfully without signs of graft-dysfunction due to assumed development of immunological tolerance.

Among patients who were alive at last time of follow-up (*n* = 14), CNIs were reduced in 50% of patients (*n* = 7), discontinued in 35.7% of patients (*n* = 5), and in 14.3% of patients CNIs remained unmodified.

Regarding patients, who deceased in the observation period, tumor diagnosis led to reduction of CNIs in 54.2% of patients and in further 2.1% to a withdrawal of CNI medication, whereas no changes in immunosuppression were pursued in 43.8% of patients (*n* = 21).

A restrictive immunosuppressive management (RIM) i.e., discontinuation or significant reduction of IS after diagnosis of lung carcinoma, was associated with a significant advantage in short- and long-term survival with median survival of 38.6 months compared to 6.7 months in patients without modification of IS (Figure 4 median survival 38.6 months (CI: 12.1–65.1; IQR 13.5) vs. 6.7 months (CI: 0.0–17.5; IQR 5.5), respectively; log rank *p* = 0.018, Breslow *p* = 0.001).

With respect to reduction of CNI, a statistically significant prolongation of median survival was observed compared to patients whose IS levels were not reduced (Figure 5 median survival in patients with reduction of CNI 38.6 months (CI: 14.8–62.3; IQR 12.1) vs. median survival in patients with withdrawal of CNI 18.9 months (CI: 13.7–24.1; IQR 2.7) vs. median survival in patients with no reduction of CNI 6.7 months (CI: 0.0–17.5; IQR 5.5), respectively; log rank *p* = 0.041, Breslow *p* = 0.003). This positive effect on survival rates was also evident in patients being weaned of CNI therapy. Withdrawal of IS was performed in six patients of which five LT recipients are still alive and resulted in a survival benefit compared to patients without reduction of CNI dosages (Figure 5). 

The significant prolongation of survival rates following CNI reduction was also evident in LT recipients who did not meet the criteria for surgical resection. In patients who did not qualify for surgery, minimalization of IS was associated with a significant improvement of survival rates compared to no reduction of IS (Figure 6 median survival 13.2 months (CI: 0.0–30.7; IQR 8.9) vs. 3 months (CI: 1.2–4.8; IQR 0.9), respectively; log rank *p* = 0.032, Breslow *p* = 0.008).

However, conversion to mTOR with mean dosage of 1.5 mg per day and median trough level of 2.4 ng/mL (range 1.4–6.2 ng/mL) did not show a benefit in survival rates. Median survival following switch to mTOR was 39.0 months (CI: 6.9–71.1; IQR 16.4) compared to 17.8 months (CI: 7.5–28.1; IQR 5.2) in case of renunciation of mTOR (Figure 7 log rank *p* = 0.756, Breslow *p* = 0.494).

MMF used as a part of IS regimen after diagnosis of de novo lung carcinoma in 17 patients with mean dosage of 1.1 g per day and a median MMF trough level of 1.6 µg/mL (range 0.5–2.14 µg/mL) did not seem to affect median survival after tumor diagnosis compared to 45 patients who did not receive MMF (Figure 8 median survival 20.2 months (CI: 6.4–34.1; IQR 7.1) vs. 19.0 months (CI: 6.5–31.4; IQR 6.3), respectively; log rank *p* = 0.764; Breslow *p* = 0.954). The effect of MMF comedication showed no significant correlation on the time between LT and tumor diagnosis.

In total, acute cellular rejection (ACR) was present in 30.6% (*n* = 19) of the study population (Table 1). The majority of ACR occurred prior diagnosis of de novo lung carcinoma (*n* = 13 of 19; 68.4%) and six patients (*n* = 6 of 19; 31.6%) developed ACR thereafter. Further analysis of the reduction of IS therapy in these six patents with presence of ACR after diagnosis of lung cancer showed that CNI dosage was reduced from a mean dosage of 2.3 mg per day and median trough level of 5.0 ng/mL (range 3–10.5 ng/mL) to mean dosage of 1.3 mg per day and median trough level of 1.5 ng/mL (range 0–3 ng/mL). Of note, no graft loss occurred.

We performed Cox-regression analysis of potential confounders (Table 3). In accordance, LT recipients who underwent surgical therapy showed a significant survival benefit compared to patients without surgical approach, with a hazard ratio of 0.249 (CI: 0.126–0.494; *p <* 0.0001). Besides oncological treatment mode, CNI-reduction including withdrawal remained the most significant factor, with a hazard ratio of 0.460 in Cox-regression analysis (CI: 0.232–0.912; *p* = 0.026).

## 4. Discussion

In this retrospective analysis from a large volume single transplantation center, de novo lung carcinoma was diagnosed in 62 LT recipients and thus, identified as a severe adverse event following LT.

Since cumulative exposure to IS is associated with development of de novo malignancies [5,11,12], lung cancer in LT recipients usually occurs years after transplantation with time of presentation frequently ranging between two and six years after LT [4,15,17]. With a median follow up of 9.7 years after LT, patients at our transplantation center were diagnosed in a later point of time than reported in other studies. This circumstance might be attributed to the sparse and individual IS medication according to our institutional standard operating procedure.

Alcohol-related cirrhosis was identified as the main indication for LT among liver recipients developing de novo lung carcinoma. This effect might be attributed to the epidemiological association between alcohol abuse and smoking resulting in higher rates of lung cancer among patients who underwent LT for alcohol-related cirrhosis compared to other indications [15,18].

In consistence with previous studies, mean age at time of diagnosis was 62.8 years [17] and NSCLC was observed most commonly with a disproportionately high frequency of squamous cell carcinoma compared to adenocarcinoma [4]. Palliative treatment due to advanced stages was associated with limited prognosis with a median survival rate of 6.4 months after tumor diagnosis. On the contrary, recipients qualifying for the surgical approach benefited from a significantly improved long-term survival, underlining the importance of early tumor detection as part of the aftercare programs following LT. Even though an increase in 90-day postoperative mortality and rate of postoperative infection was demonstrated in a recent observational study, long-term survival was similar to that of the general population [19]. Moreover, in cox-regression analysis of potential confounders on survival rates surgical approach was identified as the most significant factor. These results underline the importance of surgical treatment and as a precondition for this, the diagnosis at an early stage.

Thus, surgical treatment if technically feasible should be offered to immunosuppressed transplant recipients as a curative therapeutic option in patients with earlier-stage de novo lung carcinoma.

Immunosuppressive pharmacotherapy following LT is characterized by a personalized configuration of substances, dosage, and drug combination aiming to suppress alloimmune responses while taking each patient’s etiology of primary liver disease, comorbidities, and metabolism into account [14,20]. The backbone of IS therapy during maintenance phase is based on CNI with a focus on tacrolimus as well as MMF and mTORi [2,14,20].

Moreover, the contributing effect of IS on oncogenesis is well established as the loss of immunological integrity caused by IS results in impaired tumor recognition [13,14,21]. Despite the hypothesis that the carciogenic effect of IS therapy is dose-related and enhanced with prolonged exposure to IS, guidelines addressing the management of IS after diagnosis of de novo malignancy after LT are missing to this date [4,9,13].

In present days, a raising trend towards lowering drug levels of tacrolimus-based IS in LT recipients with stable graft function is already noticeable to avoid infections, nephrotoxicity, and malignancies as direct outcome of IS [14]. Based on previous studies demonstrating the feasibility of RIM in selected patients with long-term stable liver function and without history of autoimmune disease [14,22], the diagnosis of de novo lung carcinoma initiated a modification of IS in terms of reduction of CNI in 53.2% of our study population. Reduction of CNI was accomplished according to our institutional standard operating procedure based on close surveillance in an interdisciplinary approach. RIM was initially performed with the attempt to reduce CNI-related toxicities but interestingly contributed to a significant survival benefit of 32 months compared with patients without alteration of IS following diagnosis of lung carcinoma. Graft failure or major side effects were not observed in this subgroup.

Thus, our findings suggest that further individualization of IS for selected recipients with diagnosis of de novo lung cancer is mandatory to improve limited survival and that reduction of CNI to even cessation qualifies as a feasible treatment strategy in these cases.

Three decades ago, the possibility to exploit reduction of life-long IS therapy to an extent of complete withdrawal of IS was originally introduced and is since then, subject of discussion [23]. Collective clinical experiences demonstrated that permanent discontinuation of IS while retaining stable graft function can be attempted safely in 19.4% of LT recipients [24,25]. This effect is attributed to development of operational tolerance [25,26,27].

In an European prospective randomized multicenter trial referred to as the Immune Tolerance Network ITN030ST A-WISH trial (NCT00135694) complete withdrawal of IS was attained successfully in over 40% of highly selected liver transplant recipients due to presumed operational tolerance [28]. Interestingly, patients who presented with acute rejection in the course of IS discontinuation were not exposed to increased risk of loss of allograft function once resumption to baseline IS with or without additional administration of steroid bolus was established [24]. Thus, the persistent concern that absence of IS poses a harmful threat towards allograft function has been disproved. In our study, an elevated incidence of acute rejection following IS weaning was not evident. In fact, our findings suggest that minimalization of IS to the point of complete IS discontinuation poses no increased risk for rejection. Notably, increased time with a mean of 10.6 years since LT emerged as the key predictor for successful withdrawal [28].

mTORi are immunosuppressive agents mainly reserved to recipients with identified risk factors for post-transplant malignancies or tumor diagnosis. Conversion to mTORi is based on its potential antiangiogenetic and antiproliferative properties along with its synergistic effects towards cellular apoptosis which is sufficient reason for its usage as an oncological cotreatment during chemotherapy. Clinically, there is some evidence that mTORi have a protective effect against HCC recurrence and that in recipients with HCC a benefit in overall survival and recurrence-free survival is noticeable compared to CNI-based IS [1,13]. With respect to other post-transplant malignancies, mTORi are also possibly beneficial; however, our retrospective study did not show an improvement in survival of recipients whose IS therapy was converted to mTORi. The main positive effect of IS modification cannot be attributed to the introduction of mTORi but to the reduction of CNI.

Our retrospective study is subject to several limitations. The findings of our study are limited due to the retrospective character of the study and patients who received diagnosis and treatment in an external clinic. All data were assessed from a single transplantation center with an appropriate study population being obtained over the years. However, uninterrupted follow-up was impaired by continuously increasing patient numbers due to far-reaching outpatient treatment and extremely long observation periods of almost 30 years.

Furthermore, potential confounders such as physical constitution, medical history prior to diagnosis of malignancy, and epidemiological associations should be acknowledged. In our study, secondary endpoints, such as quality of life, were not assessed. On the other hand, the results of the analysis show long-term phenomena after LT according to the real world scenario of a transplant center with a 30-year experience.

Another aspect that should be taken into account is the heterogeneity in IS dosages since evidence-based protocols regarding the modification and reduction of IS are still missing to the present day. Thus, RIM was defined as any IS reduction compared to the dosage prior to cancer diagnosis without usage of specific thresholds. Further investigations concerning dosage findings and reduction of IS in organ recipients should be the targets of subsequent studies.

## 5. Conclusions

In case of diagnosis of de novo lung carcinoma reduction of IS to the extent of withdrawal should be taken into account to minimize complications and toxicities resulting from prolonged exposure to IS. The risk of acute rejection does not appear to be increased with RIM. Modification of IS should always be performed in balance with the risk of allograft rejection and should be guided by histopathological evaluation as gold standard.

LT recipients with the diagnosis of de novo lung cancer should be offered surgical treatment if technically feasible as a potential curative therapeutic option to improve limited prognosis.

The results of our study justify further research to establish evidence-based guidelines on reduction of IS and to elucidate the applicability and role of IS withdrawal prior development of posttransplant malignancies.

## Figures and Tables

**Figure 1 cancers-14-02748-f001:**
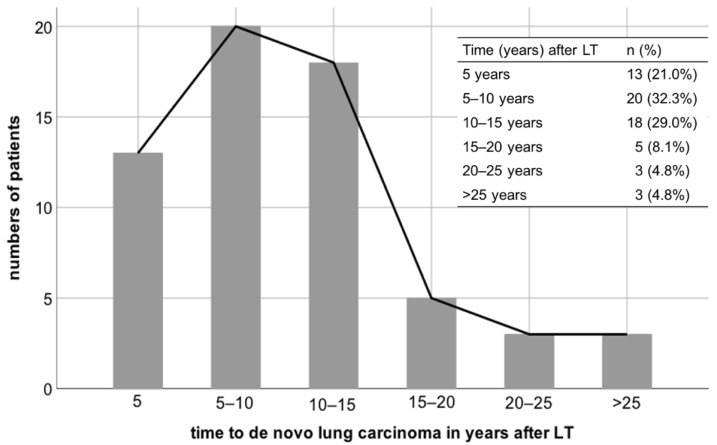
Time of diagnosis of de novo lung carcinoma after liver transplantation. LT—liver transplantation.

**Figure 2 cancers-14-02748-f002:**
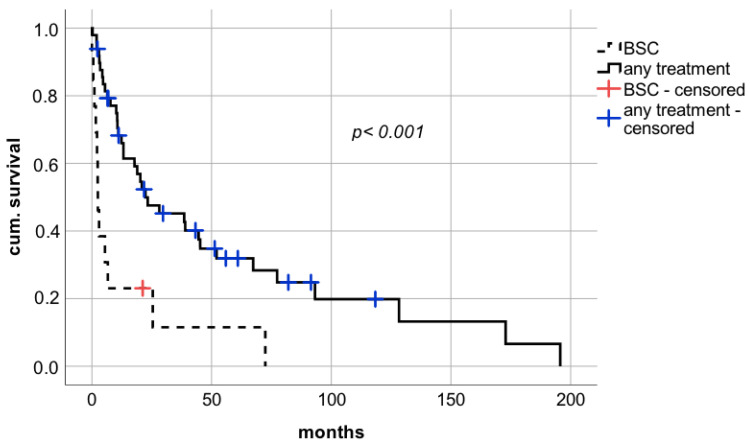
Impact of therapeutical strategies on survival diagnosis of de novo lung carcinoma after liver transplantation. BSC—best supportive care.

**Figure 3 cancers-14-02748-f003:**
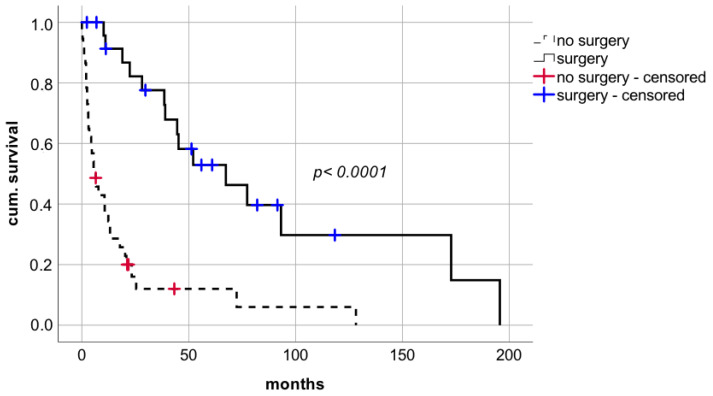
Comparison of patients with or without surgical approach after diagnosis of de novo lung carcinoma after liver transplantation.

**Figure 4 cancers-14-02748-f004:**
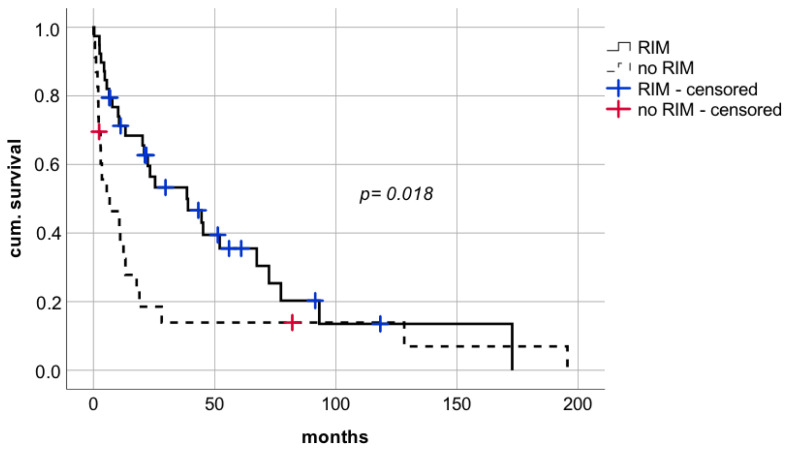
Comparison of patients with or without restrictive immunosuppressive management after diagnosis of de novo lung carcinoma after liver transplantation. RIM—restrictive immunosuppressive management.

**Figure 5 cancers-14-02748-f005:**
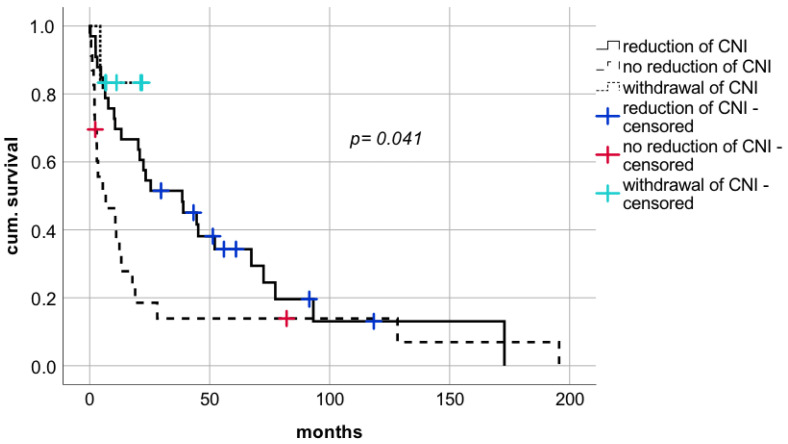
Impact on modification of immunosuppressive therapy after diagnosis of de novo lung carcinoma after liver transplantation. CNI—calcineurin inhibitors.

**Figure 6 cancers-14-02748-f006:**
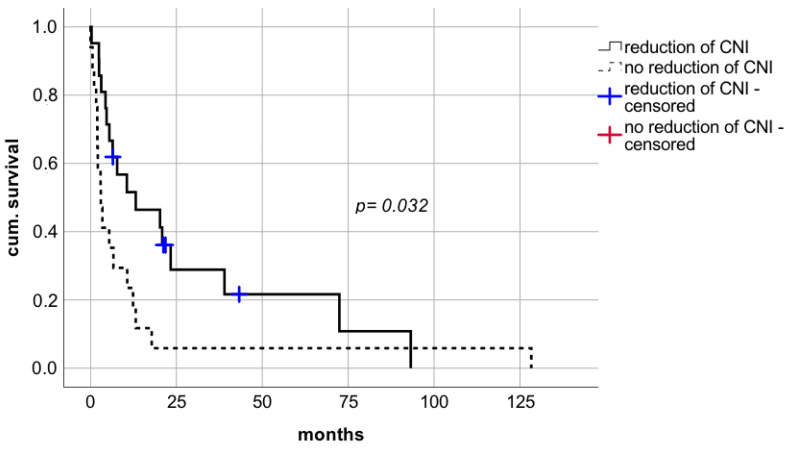
Impact on modification of immunosuppressive therapy in patients with palliative treatment after diagnosis of de novo lung carcinoma after liver transplantation. CNI—calcineurin inhibitors.

**Figure 7 cancers-14-02748-f007:**
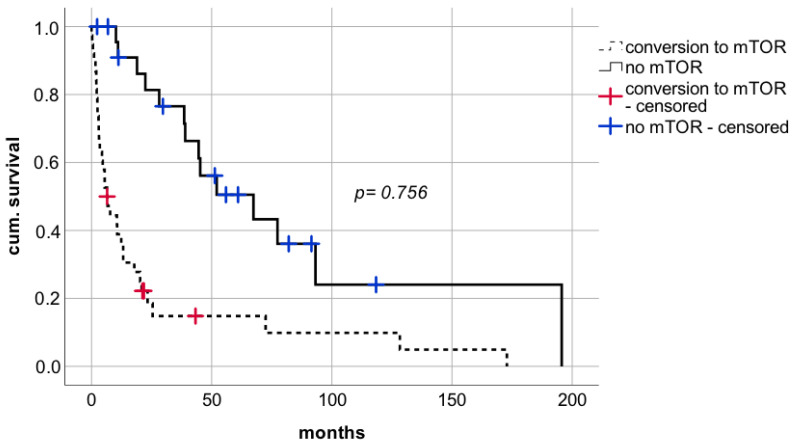
Impact on conversion to mTOR after diagnosis of de novo lung carcinoma after liver transplantation. mTORi—mammalian target of rapamycin inhibitors.

**Figure 8 cancers-14-02748-f008:**
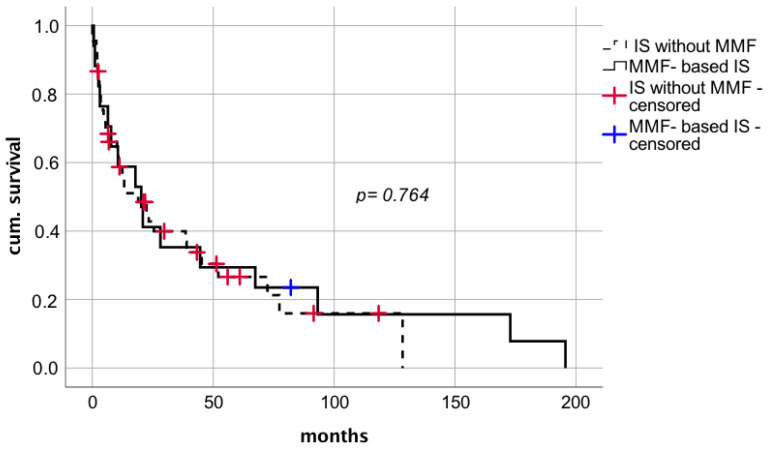
Impact on MMF-based IS after diagnosis of de novo lung carcinoma after liver transplantation. IS—immunosuppression; MMF—mycophenolate mofetil.

**Table 1 cancers-14-02748-t001:** Patient characteristics.

All Patients with De Novo Lung Cancer after LT	*n* = 62
Sex (%)	
male	36 (58.1)
female	26 (41.9)
Median age at LT in years (range)	53.5 (31.3–65.5)
Indication for LT (%)	
ALD	29 (46.8)
HCV	10 (16.1)
HBV	12 (19.4)
AIH, PBC, PSC	6 (9.7)
others	5 (8.1)
HCC (%)	12 (19.4)
Retransplantation (%)	6 (9.7)
Median age at de novo lung cancer in years (range)	62.8 (45.7–81.0)
Median time to lung carcinoma in years (range)	9.7 (0.7–27.0)
Median survival in months (range)	13.2 (0–196)
IS regimen at time of diagnosis (*n* = 62)	
tacrolimus mono (%)	30 (48.4)
tacrolimus plus prednisolon (%)	1 (1.6)
tacrolimus plus MMF (%)	15 (24.2)
CsA mono (%)	8 (12.9)
CsA plus MMF (%)	4 (6.5)
tacrolimus extended release (%)	2 (3.2)
MMF mono (%)	2 (3.2)
ACR (%)	
prior tumor diagnosis (%)	13 (20.9)
post tumor diagnosis (%)	6 (9.7)
total (%)	19 (30.6)
Status at last follow-up (*n* = 62)	
Alive (%)	14 (22.6)
Deceased (%)	48 (77.4)

LT—liver transplantation; ALD—alcoholic liver disease; HCV—hepatitis C virus; HBV—hepatitis B virus; AIH—autoimmune hepatitis; PBC—primary biliary cirrhosis; PSC—primary sclerosing cholangitis; HCC—hepatocellular carcinoma; IS—immunosuppression; MMF—mycophenolate mofetil; CsA—Cyclosporine A; mTORi—mTOR inhibitor; ACR—acute cellular rejection.

**Table 2 cancers-14-02748-t002:** Tumor characteristics and treatment.

Characteristics	All Patients *n* = 62	Alive *n* = 14	Deceased *n* = 48
Tumor entity (*n* = 62)			
NSCLC (%)	53 (85.5)	14 (100.0)	39 (81.3)
*Squamous cell carcinoma (%)*	*27 (50.9)*	*7 (50.0)*	*20 (51.3)*
*Adenocarcinoma (%)*	*24 (45.3)*	*7 (50.0)*	*17 (43.6)*
*Others (%)*	*2 (3.8)*	*0 (0)*	*2 (5.6)*
SCLC (%)	5 (8.1)	0 (0)	5 (10.4)
Unknown (%)	4 (6.5)	0 (0)	4 (8.3)
TNM stage (*n* = 62)			
stage I (%)	20 (32.3)	9 (64.3)	11 (22.9)
stage II (%)	4 (6.4)	1 (7.3)	3 (6.3)
stage III (%)	6 (9.7)	2 (14.3)	4 (8.4)
stage IV (%)	7 (11.3)	1 (7.1)	6 (12.5)
Unknown (%)	25 (40.3)	1 (7.1)	24 (50.0)
Oncologic regimen (*n* = 62)			
BSC (%)	13 (21.0)	1 (7.1)	12 (25.0)
chemotherapy/radiotherapy (%)	24 (38.7)	3 (21.4)	21 (43.8)
surgery alone (%)	19 (30.6)	8 (57.1)	11 (22.9)
surgery and chemotherapy (%)	6 (9.7)	2 (14.3)	4 (8.3)
Modification of IS (*n* = 62)			
reduction of CNI (%)	33 (53.2)	7 (50.0)	26 (54.2)
no reduction of CNI (%)	23 (37.1)	2 (14.3)	21 (43.8)
withdrawal of CNI (%)	6 (9.7)	5 (35.7)	1 (2.1)
mTOR inhibitor (%)	9 (14.5)	3 (21.4)	6 (12.5)

NSCLC—non-small cell lung cancer; SCLC—small cell lung cancer; TNM—TNM Classification of Malignant Tumors; BSC—best supportive care; IS—immunosuppression; CNI—calcineurin inhibitor; mTORi—mTOR inhibitor.

**Table 3 cancers-14-02748-t003:** Potential confounders.

Parameters	*p*	Hazard Ratio	95% CI	
			Lower	Upper
Age	0.257	1.026	0.981	1.073
Gender	0.681	0.870	0.447	1.693
Tumor entity	0.193	1.283	0.881	1.869
Surgery	<0.001	0.249	0.126	0.494
CNI-reduction	0.026	0.460	0.232	0.912

CI—confidence interval; CNI—calcineurin inhibitor.

## Data Availability

The data presented in this study are available on request from the corresponding author. The data are not publicly available due to conditions of the ethics committee of our university.

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
