# Peer review of "Reducing Immunosuppression in Patients with De Novo Lung Carcinoma after Liver Transplantation Could Significantly Prolong Survival"

_cancers, 2022, doi:10.3390/cancers14112748_

Round 1

Reviewer 1 Report

This work is very interesting but lack of some important information.

1.Doses and blood concentrations of IS before and after of drug reduction. Did the extent of drug reduction impact the results?

2.Tumor characteristics in three groups must be provided, cause the survival  differences were much larger than my past knowledge.

3.The IS redcution and waening seems did not increase ACR significantly, which is very interesting. The detailed protocol of IS reduction and weaning would be appreciated. 

Reviewer 2 Report

The median survival is 10 months ( 0 - 197 ), is possible to divide it in the time to de novo , for example before 5 Y , 5 to10 y , 10 to 20 , and after 20 y from the transplant ?

Could you explain why there is a so big excursion in time for median survival ?

Could you also explain which is the protocol of follow up about CT scan of the lung in particular after transplant and how was the treatment for  early de novo, less than 5 or 5 to 10 , probably because was a cancer aggressive?

Reviewer 3 Report

Pesthy S. analyzed the impact of the reduction of IS for the lung cancer patients after LT. This manuscript has some messages to the readers of the cancers, but some refinements are needed for this manuscript.

1)    It is crucial how the immunosuppressants were reduced after the diagnosis of lung cancer. Please show how much and how about the duration.

2)    Were abdominal ultrasound performed only at 1,3,5,7,.. years? Not every 3 months?

3)    In table 1, “MMF plus CNI”, “MMF plus CsA” are confusing. Please show CsA and Tac, separately.

4)    Figure 1 should be deleted.

Round 2

Reviewer 1 Report

Conclusion: "AR"(acute rejection) is no obviously increase with the reduction of IS dosages or complete withdrawal IS. 

Author Response

We are grateful for the valuable comments received from you during the revision process. 

As extensive editing of English language and style has been requested by you, we are pleased to inform you that grammar, spelling, punctuation and phrasing of our manuscript has been rechecked thoroughly to meet the demands. Please find our revised manuscript in the attachment. 

In accordance with your remark on acute rejection and reduction of IS, we have added to the conclusion that the risk of acute rejection does not appear to be increased with restrictive IS management. 
